# Pharmacological Interactions in the Elderly

**DOI:** 10.3390/medicina56070320

**Published:** 2020-06-28

**Authors:** Emilia Błeszyńska, Łukasz Wierucki, Tomasz Zdrojewski, Marcin Renke

**Affiliations:** 1Department of Occupational, Metabolic and Internal Diseases, Medical University of Gdańsk, 81-519 Gdynia, Poland; marcin.renke@gumed.edu.pl; 2Department of Preventive Medicine & Education, Medical University of Gdańsk, 80-210 Gdańsk, Poland; lukasz.wierucki@gumed.edu.pl (L.W.); tz@gumed.edu.pl (T.Z.)

**Keywords:** geriatrics, aging, drug interactions, medication errors/prevention and control, polypharmacy, multimorbidity

## Abstract

Pharmacological therapy in the elderly is particularly complicated and challenging. Due to coexistence of three main predisposing factors (advanced age, multiple morbidity and polypharmacotherapy), this group of patients is prone to occurrence of drug interactions and adverse effects of incorrect drug combinations. Since many years patient safety during the treatment process has been one of key elements for proper functioning of healthcare systems around the world, thus different preventive measures have been undertaken in order to counteract factors adversely affecting the therapeutic effect. One of the avoidable medical errors is pharmacological interactions. According to estimates, one in six elderly patients may be at risk of a significant drug interaction. Hence the knowledge about mechanisms and causes of drug interactions in the elderly, as well as consequences of their occurrence are crucial for planning the process of pharmacotherapy. For the purpose of pharmacovigilance, a review of available methods and tools gives an insight into possible ways of preventing drug interactions. Additionally, recognizing the actual scale of this phenomenon in geriatric population around the world emphasizes the importance of a joint effort among medical community to improve quality of pharmacotherapy.

## 1. Introduction

Patient safety during the treatment process is one of key elements for proper functioning of healthcare systems around the world. An increasing emphasis is being placed on preventive measures to counteract the occurrence of events adversely affecting the therapeutic effect. One of the avoidable medical errors is pharmacological interactions, defined as an interaction of two drugs that can lead to a quantitative and/or qualitative change in the action of one of them [1]. Undesirable pharmacological interactions may increase toxicity of a drug or reduce its effectiveness.

Predisposing factors for the occurrence of adverse effects of incorrect drug combinations are primarily advanced age, multiple morbidity and polypharmacotherapy [2]. It is estimated that in developed countries about 30–40% of people over 65 years old take 5 or more drugs, while 12% of patients in this age group use 10 or more different medicines [2]. Patients burdened with the coexistence of several chronic diseases have a greater risk of drug interactions due to complexity of the treatment process and more frequent contacts with various representatives of the healthcare system [3]. Pharmacological therapy in the elderly is particularly complicated due to age-related changes in pharmacokinetic and pharmacodynamic metabolism of drugs that increase or decrease sensitivity of these patients to chemical substances [4].

A significant increase in life expectancy in the 20th century is considered to be one of the greatest social achievements. In combination with a decreasing fertility rate, it leads to a progressive aging of the population. The population over 65 years old is expected to increase from 524 million in 2010 (8 percent of the world population) to 1.5 billion (16 percent of the world population) in 2050. This increase will be mostly pronounced in developing countries [5].

Understanding the basis of drug interactions in the elderly and the consequences of their occurrence, as well as knowledge of the actual scale of this phenomenon in the population and available methods of side effects prevention, can help optimize the process of pharmacotherapy, and thus increase safety of elderly patients.

## 2. Metabolism in the Old Age

The aging process is characterized by significant changes in body composition and physiological decline in the function of most organs. With age, total body fat mass increases, water content decreases, which in combination with a reduction in muscle mass can lead to sarcopenia and changes in distribution of drugs [6]. It is estimated that body fat content over the age of 70 is about 25% in men and 40% in women, which is 1.5 times more than in young people aged 20–29 [7]. As a result, distribution volume of lipophilic drugs increases in elderly patients. Aging is also associated with a progressive decrease in body water content, which causes a decrease in distribution volume of hydrophilic compounds [8]. From early adulthood up to the age of 79–80 men present a gradual decline in total body water of 0.3 kg per year, while in women total body water remains relatively constant from youth up to the age of 70, when there is a decrease of 0.7 kg per year [9].

The most important pharmacokinetic changes include decreased renal secretory capacity and impaired hepatic metabolism. Due to the decrease in the secretory capacity of kidneys, removal of hydrophilic substances through kidneys may be significantly impaired and requires adjustment of drug dose to renal parameters [10]. Changes in liver during its aging may affect drug metabolism in many ways. Size of liver decreases with age by 20–40%, while blood flow through this organ decreases by 40–60%. In people over 70 years old, cytochrome P450 (CYP450) oxidase activity may decrease by up to 30% compared to young adults [11].

In addition, plasma protein concentration changes with age, including albumin and alpha-1 acid glycoprotein, the main roles of which are transport and storage of most exogenous and endogenous substances in bloodstream [12]. The level of albumin in the elderly is on average 19% lower than in young population. An increased free drug concentration of substances strongly bound to albumin may be a potential cause of toxicity, even if a total drug concentration is within a therapeutic range [13]. A slight increase in alpha-1 acid glycoprotein, observed in healthy elderly, but less pronounced in the presence of a disease, may cause an increase in the concentration of some substances, e.g., propranolol [14].

Studies have also shown lower levels of p-glycoprotein concentration in brain and intestinal tissue of elderly people, especially in the presence of dementia [15,16]. This membrane protein transports various substrates across the cell membrane including drugs, such as steroids or cardiac glycosides like digoxin.

Another important change observed in elderly is a decrease in cellular metabolism, which can significantly reduce the activity of substances administered in the form of prodrugs [17].

Pharmacodynamic changes are also common and are associated with changes in drug sensitivity, regardless of chemical compounds distribution in tissues [10,11]. This is illustrated by a decrease in cardiovascular sensitivity of the elderly to agonists and antagonists of beta-adrenoreceptors and an increase in frequency of orthostatic hypotension episodes after antihypertensive drugs. 

Moreover, central nervous system of elderly patients is becoming more susceptible to agents affecting brain function, e.g., opioids, benzodiazepines and psychotropic drugs [6].

## 3. Multimorbidity

Multimorbidity is a co-occurrence of at least two chronic diseases that cannot be cured at the current state of knowledge, but it is possible to control them by means of pharmacology or other therapeutic methods. According to estimates, the incidence of this phenomenon in older people is 65–98% [18,19,20] and is gradually increasing due to a better quality of healthcare and aging of the population [21,22]. Nearly half of people over 65 years old have at least three chronic diseases, while one in five people is affected by five or more chronic diseases [23,24]. The most common are hypertension, osteoarthritis, ischemic heart disease and diabetes [25].

By 2035, it is prognosed that the number of people living with two or more chronic conditions will increase by 86.4%—with the biggest rises observed for cancer (increase by 179.4%) and diabetes (increase by 118.1%) [26].

Multimorbidity in the elderly can impair drugs handling and administration. This concerns especially chronic conditions like dementia, Parkinson’s disease, stroke, esophageal diseases (e.g., gastro-esophageal reflux disease), the presence of visual or cognitive impairments and swallowing difficulties [27].

Compared with people burdened with one chronic disease, patients with multiple diseases have an increased risk of functional impairment [28,29], deterioration in quality of life [30,31] and increased mortality [32,33]. Due to the coexistence of many chronic diseases, the elderly are the main consumers of both prescription and OTC (over-the-counter) drugs, buying 15% more than they did about 10 years ago [24].

## 4. Polypharmacotherapy

Polypharmacotherapy is traditionally defined as a long-term use of five or more drugs. It is estimated that in developed countries about 30–40% of people over 65 years old take 5 or more drugs, while 12% of patients in this age group use 10 or more different medicines [2]. The report of the World Health Organization in 2019 confirms that polypharmacotherapy is a common problem, but diverse structures of healthcare provision and different information collection systems make it difficult to compare data from individual countries [34].

In addition, in recent years there has been a growing interest among older people in dietary supplements, which are widely available on the pharmaceutical market. It is estimated that over 40% of people over 60 years old declare to consume products from this group, and the value of dietary supplements market in the European Union is estimated at five billion euros. The most commonly used supplements are vitamin and mineral products, fish oils, probiotics and some herbal products [35].

It is worth emphasizing the role of excipients, which are generally considered to be biologically inert or inactive. Recent research has proven they can influence a drug effect by contributing synergistically (e.g., antitumor activity and cardiovascular benefits of gamma-linolenic acid and omega-3 fatty acids) or by reducing drug effectiveness through different mechanism (e.g., degradation of neomycine by sodium carboxymethylcellulose or absorption of nitrazepam in tablet dosage form by colloidal silicon) [36]. Excipients can also alternate (decrease or increase) the permeability of intestinal membrane, and therefore affect bioavailability of the drug [37]. Due to age-related changes in metabolism, accumulation of excipients may occur in older patients, leading to toxicity and adverse effects. A list of such excipients are summarized in a review [38].

The literature provides a broad and accurate description of negative medical, economic and social consequences of polypharmacotherapy. The most dangerous are drug interactions, cognitive impairment [23], weight loss and malnutrition [39], falls and bone fractures [40], rehospitalization [41], reduced quality of life [42] and death [42].

Poor nutritional status and malnutrition in the elderly population are important areas of concern, associated with an increased financial cost burden [43] and reduced quality of life [44]. Drug classes with potentially significant drug–nutrient interactions include: antihypertensives by causing zinc deficiency (thiazide diuretics, angiotensin receptor blockers, angiotensin converting enzyme inhibitors and potassium-sparing diuretics); acetylcholinesterase inhibitors; proton pump inhibitors and metformin by causing vitamin B12 deficiency; HMG-Co reductase inhibitors (statins) by causing reductions in CoQ10, a-tocopherol, b-carotene and lycopene and long-term, high-dose aspirin by causing vitamin C deficiency [45]. Additionally, polypharmacy may negatively affect the therapeutic success of vitamin D supplementation by reduced compliance and adherence to treatment, and interference with drug adsorption, revealing the need for a higher amount of vitamin D supplements [46].

Drugs frequently consumed by the elderly (including antihypertensives, hypolipemic, hypoglycemics or drugs for acid- or nervous-related disorders) may trigger hypohydration by means of increased water elimination (through diarrhea, urine or sweat); decreased thirst sensation or appetite or alteration of central thermoregulation. On the other hand, excipients induce alterations in hydration status by decreasing gastrointestinal transit time or increasing the gastrointestinal tract rate or intestinal permeability [47].

Studies show that, on average, 1 in 5 commonly used drugs by the elderly should not be recommended [48], while for people in nursing homes and other institutions this problem may affect up to 30–50% of patients [49]. According to estimates, one in six elderly patients may be at risk of a significant pharmacological interaction [24].

## 5. Prevalence of Drug Interactions

The phenomena of polypharmacotherapy and pharmacological interactions have been intensively analyzed in Western Europe and the United States, while data from Central and Eastern European countries are still limited. Most of the analyses were based on screening tool of older persons’ potentially inappropriate prescriptions (STOPP)/screening tool to alert doctors to the right treatment (START) or Beers criteria and concerned outpatient patients.

A research conducted in 2009 on nearly 1300 Northern Ireland citizens over the age of 65 showed that 18.3% of them used dangerous drug combinations [50]. A comprehensive study conducted in the United Kingdom in 2003 included 131 primary care outpatient clinics with approximately 162,000 patients over 65 years old annually—24.8% of them were shown to use abnormal drug combinations, while 20.5% of patients took high-risk drugs [51]. A nationwide survey conducted in Lithuania in 2017 on over 400,000 citizens over the age of 65 years old showed a higher percentage of incorrect drug combinations, of around 25% [52]. A similar result was obtained in 2005 in Portugal, where more than 25% of 213 examined patients over 65 years old used at least one drug not indicated for use in the elderly [53]. In some countries lower values of incorrect drug connections have been reported in the elderly, e.g., 9.8% in Turkey [54] or 12.5% among Finnish citizens [55].

So far, a study was conducted in Poland in 2007 on 1000 seniors from Poznań and Głogów, finding the percentage of incorrect drug combinations of 28.6% [56]. This relatively high rate may be due to consumption of OTC drugs that accounted for 5.5% of incorrect connections. Unfortunately, there is a lack of national data determining the incidence of pharmacological interactions in Polish elderly patients.

The percentage of incorrect drug combinations in European studies ranges between 9.8 and 38.5% and is relatively higher than in the United States (21.3–28.8%) [57,58,59,60]. The reasons for the observed discrepancies are unclear and may be due to differences in the availability of drugs in individual countries, different practices of prescribing drugs or verifying prescriptions by pharmacists.

## 6. Consequences of Pharmacological Interactions

### 6.1. Impact of Pharmacological Interactions on Life Expectancy

The most serious effect of adverse reactions due to pharmacological interactions is death [61]. Studies in the United States have shown a 1.6-fold increase in mortality in people taking abnormal drug combinations [62]. Importantly, inclusion of another maladjusted drug is associated with a relative increase in mortality by 39%, regardless of the number of medicines [63]. Particularly strong association was observed between an increase in mortality and incorrect combinations with drugs rising the risk of falls in elderly (STOPP criteria version 2, category K), e.g., benzodiazepines, neuroleptics, hypnotics from the Z group [64].

### 6.2. Impact of Pharmacological Interactions on Quality of Life

Definition of health-related quality of life (HRQoL) includes a subjective perception of the ability to perform activities that are important to a person and are affected by current health [65]. In previous studies, association between a large number of drugs taken, potential pharmacological interactions and reduced quality of life has been observed [66]. These patients had significantly reduced health-related quality of life self-assessment scores on the EQ-VAS scale (mean 63.12 ± 17.37 points) [67]. This effect was intensified by an accompanying deterioration of mobility, functional fitness and cognitive functions [67,68].

### 6.3. Impact of Pharmacological Interactions on the Frequency of Rehospitalization

Adverse reactions due to pharmacological interactions are among the most common causes of hospitalization in the elderly [69]. In a nationwide observational study in the United States, the incidence of sudden hospitalizations due to adverse drug reactions was 99,628 cases per year [70]. For the same reason, in 2005 the number of visits to various health care institutions was around 4.3 million [71]. It is estimated that in two out of three cases rehospitalization could be avoided [69].

### 6.4. Impact of Pharmacological Interactions on Financial Costs

In the United States the cost of medical treatment due to medical errors and side effects in the elderly population is estimated at over $200 billion a year [72]. In European countries this phenomenon is also a significant problem [73,74,75]. Frequent visits to emergency departments and outpatient facilities, rehospitalizations, as well as purchase of many pharmacological products are a financial burden for individual patients and the entire healthcare system [2].

## 7. Prevention of Drug Interactions

Due to the research carried out by specialists in the field of gerontology and geriatrics, the knowledge about mechanisms of the aging process is growing. Through further analyses, the main factors predisposing to the occurrence of pharmacological interactions have been recognized and it is more probable to identify patients who are particularly at risk of drug side effects. In addition, there is an awareness of the consequences of pharmacological interactions, which not only affect health and quality of life of patients, but also disrupt the therapeutic process and constitute a financial burden on health care systems.

Previous studies show that 80% of serious drug side effects are caused by an incorrect prescription. Moreover, on average 87.9% of these events are potentially predictable and avoidable [61].

A variety of standardized tools are available to assist in the planning of pharmacotherapy according to individual needs and capabilities of older people. Some of them contain concise guidelines for recommended and contraindicated drugs in geriatric patients, e.g., Beers criteria, STOPP criteria (screening tool of older persons’ potentially inappropriate prescriptions) and START (screening tool to alert doctors to the right treatment). Their effectiveness has been extensively studied in many countries [76], indicating the predominance of STOPP/START criteria in identifying inadvisable drugs [77]. In addition, unlike Beers’ criteria, STOPP/START criteria were created for usage as a checklist, which means that they are often applied in clinical work and as a part of research protocols [2]. Some countries have developed tools adapted to specifics of its geriatric population and availability of drugs on a given market, such as the Norwegian NORGEP criteria (The Norwegian General Practice) [78].

The role of a clinical pharmacist is also emphasized, whose actions can identify and minimize problems associated with incorrect pharmacotherapy. This solution has been successfully implemented many years ago in the United States and Great Britain, as well as in many European countries, leading to improved therapeutic effects and bringing economic benefits [79].

Automated interaction analysis systems are a solution that has a potential to increase recognition of drug interactions. Along with computerization of healthcare systems, various forms of dedicated programs are available around the world: on-line tools, applications for mobile devices, software modules or even as part of a medical information network—the SureScripts network in the United States [80]. Unfortunately, a reported low alert specificity can be a serious obstacle to an effective use of information and actual increase of patient safety during pharmacotherapy [81].

Despite extensive research and many years of experience, data from an overwhelming number of countries around the world still indicate an alarmingly high percentage of incorrect drug combinations in the elderly. What could be the reasons for this phenomenon?

Some researchers emphasize the imperfection of specific criteria for drugs not indicated in geriatric patients, assessing them as inflexible and not taking into account all factors determining a high quality individualized healthcare [59]. Older people are often excluded from randomized clinical trials that provide information on adverse effects associated with specific drugs [59]. Therefore, there is a lack of accurate data on relative risk and benefits of therapeutic measures in this group of patients [57].

Some specialists note a significant heterogeneity of the elderly population, which means that the risk–benefit ratio will vary depending on the patient’s clinical condition [57]. Multimorbidity among seniors results in the use of complex treatment regimens. Doctors may be reluctant to change a scheme that works because there is no evidence of a harmful effect of therapy [59]. In addition, drugs identified as not recommended in geriatric patients may be suitable as second-line medicines for a person who has not responded or does not tolerate the preferred agent [57]. Moreover, the cost of therapy is often an important factor in choosing a specific drug [57]. In some circumstances, the use of a medicine may be clinically justified, if the benefits outweigh the risks to the patient [57].

Most experts agree that all available support tools for identifying pharmacological interactions and abnormal drug combinations can best be used as a screening tool to identify elderly people at high risk of suboptimal pharmacotherapy, and to recognize and prioritize problem areas rather than as a final measure of quality or efficiency of healthcare [57].

## 8. Concluding Remarks

In the face of reported data on the high incidence of abnormal drug combinations and undeniable negative consequences of pharmacological interactions, it is necessary to draw attention of the medical community to the existing problem and to make a joint effort to improve quality of pharmacotherapy, thereby increasing safety of patients. So far, surveys conducted among physicians indicate their limited ability to recognize potential drug interactions, which creates a need to find solutions supporting the work of clinicians [82]. In the face of computerization of medicine, automatic interaction analysis systems are becoming an increasingly common source of information about potential drug interactions, however, they still meet with skepticism of part of the medical community [83]. Therefore, there is a need to improve and disseminate electronic sources of information on drug interactions, in order to create a better auxiliary tool in conducting a safe pharmacotherapy.

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
