# Peer review of "Pharmacological Interactions in the Elderly"

_medicina, 2020, doi:10.3390/medicina56070320_

Round 1
Reviewer 1 Report
The review is reasonably clear and well written; however, there are certain places where the text needs to be clarified (see specific comments below). Moreover, I have several comments that the authors may wish to address in order to strengthen the manuscript.
Line 38. The authors should avoid the term “medicament”. The proper name for the compounds with pharmacological activity is drugs. In any case, medicine is also correct, but it should be considered that a medicine include not only the drug but also de excipients.
Line 60. I would suggest including the differences of total body water percentage between the adults and the elderly.
Lines 62-77. The authors mention the differences observed in the elderly in CYP450, albumin and alpha-1 acid glycoprotein. What about the p-glycoprotrein? Are there any differences? Considering that it is an important protein of the cell membrane that pumps many foreign substances, including drugs, out of cells it may also have some important consequences.
Lines 99-118: In this section of the manuscript, the authors mention the fact that polypharmacy lead to malnutrition. I miss in the text information about the effect of drugs in nutrient levels, for example, it is well known that there are drugs affect vitamin D levels or other that lead to the appearance of dehydration. I strongly suggest including this information in the text.
Lines 154-160 and 167-172. Maltnutrition also affect quality of life and have impact on pharmaceutical expenses owing to the need to include in patient treatment vitamin supplements (for example vitamin D, widely prescribed in the elderly). Have you considered mentioning this in the manuscript?
Traditionally, pharmaceutical excipients have considered inert in terms of drug efficacy but currently there are available research studies that confirm that excipients modify drug effect and nutrients absorption by means of different mechanisms, including alterations in intestinal permeability. Definitely, this information will provide your manuscript with great value in terms of its novelty and originality.
Reviewer 2 Report
This review provides useful information about the risk of pharmacological therapy in the elderly, however, several changes should be done.
First of all, I think that the manuscript present different issues involved in the pharmacological interaction which are distributed in a logical order making easy its reading and understanding, however, the concept are poorly developed. In many cases, the different factor that contribute to interaction problems are just named without going in depth which makes this review lose value. Therefore, from my point of view, a deeper analyze and development of actual data should be done to complete this work.
Moreover, some parts a bit confusing. For example: in the metabolism in the old age, several changes are described, however, the writing is difficult to follow and sentences are not bound (lines 69-84).
Finally, I would suggest to add some graphic abstract, table with data ... to better understanding to the review.
Round 2
Reviewer 1 Report
Point 1: Line 38. The authors should avoid the term “medicament”. The proper name for the compounds with pharmacological activity is drugs. In any case, medicine is also correct, but it should be considered that a medicine include not only the drug but also de excipients.
Response 1: We have corrected all lines, where the term ”medicament”, ”medicine” or “medication” was used incorrectly, as follows:
- line 38.: “medicines” instead of “medicaments”
- line 74: “substances” instead of “medicaments”
- line 116: “medicines” instead of “medicaments”
- line 183: “drug” instead of “medication”
- line 184: “medicines” instead of “medications”
- line 191: “drugs” instead of “medications”
- line 220: “drugs” instead of “medicines”
- line 227: “drugs” instead of “medicines”
- line 245: “drugs” instead of “medications”
- line 252: “medicines” instead of “medicaments”
- line 255: “medicine” instead of “medication”
Thank you very much.
Point 2: Line 60. I would suggest including the differences of total body water percentage between the adults and the elderly.
Response 2: We have included information about differences in total body water content between adults and elderly in lines 61-63, as follows:
“This phenomenon is mostly observed in older women, whose bodies consist of approximately 10% less water than in young age, while no significant difference in total body water content occurs in men [9].”
It is not true than total body water of men in the elderly is not different than in the youth, this is a very strong affirmation, having in mind that total water content is highly dependent on different internal and external factors. I would suggest to remove this sentence and to include, as previously stated, the differences of total body water percentage between the adults and the elderly.
Point 3: Lines 62-77. The authors mention the differences observed in the elderly in CYP450, albumin and alpha-1 acid glycoprotein. What about the p-glycoprotrein? Are there any differences? Considering that it is an important protein of the cell membrane that pumps many foreign substances, including drugs, out of cells it may also have some important consequences.
Response 3: We have included information about differences in p-glycoprotein concentration in elderly people in lines 79-82, as follows:
“Studies have also shown lower levels of p-glycoprotein concentration in brain and intestinal tissue of elderly people, especially in presence of dementia [15,16]. This membrane protein transports various substrates across the cell membrane including drugs, such as steroids or cardiac glycosides like digoxin.”
Ok, thank you.
Point 4: Lines 99-118: In this section of the manuscript, the authors mention the fact that polypharmacy lead to malnutrition. I miss in the text information about the effect of drugs in nutrient levels, for example, it is well known that there are drugs affect vitamin D levels or other that lead to the appearance of dehydration. I strongly suggest including this information in the text.
Response 4: We have included information about effect of polypharmacy on nutrient levels (vitamin B12, vitamin D, vitamin C, zinc, CoQ10) in lines 140-149, as follows:
“Drug classes with potentially significant drug–nutrient interactions include: antihypertensives by causing zinc deficiency (thiazide diuretics, angiotensin receptor blockers, angiotensinconverting enzyme inhibitors and potassium-sparing diuretics); acetylcholinesterase inhibitors; proton pump inhibitors and metformin by causing vitamin B12 deficiency; HMG-Co reductase inhibitors (statins) by causing reductions in CoQ10, a-tocopherol, b-carotene and lycopene; long-term, high-dose aspirin by causing vitamin C deficiency [45]. Additionally, polypharmacy may negatively affect the therapeutic success of vitamin D supplementation by reduced compliance and adherence to treatment, and interference with drug adsorption, revealing the need for a higher amount of vitamin D supplements [46].”
Despite the manuscript have improved, I still miss information in terms of interactions between drugs and hydration status. In fact, this is the main adverse effects of widely used drugs in the elderly such as diuretics.
Point 5: Lines 154-160 and 167-172. Maltnutrition also affect quality of life and have impact on pharmaceutical expenses owing to the need to include in patient treatment vitamin supplements (for example vitamin D, widely prescribed in the elderly). Have you considered mentioning this in the manuscript?
Response 5: We have included information about effect of malnutrition on quality of life and financial expenses in lines 139-140, as follows:
“Poor nutritional status and malnutrition in the elderly population are important areas of concern, associated with an increased financial cost burden [43] and reduced quality of life [44].”
Thank you.
Point 6: Traditionally, pharmaceutical excipients have considered inert in terms of drug efficacy but currently there are available research studies that confirm that excipients modify drug effect and nutrients absorption by means of different mechanisms, including alterations in intestinal permeability. Definitely, this information will provide your manuscript with great value in terms of its novelty and originality.
Response 6: We have included information about effect of excipients on drug effect in lines 126-134:
“It is worth to emphasise the role of excipients, which are generally considered to be biologically inert or inactive. Recent research has proven they can influence drug effect by contributing synergistically (e.g. antitumor activity and cardiovascular benefits of gamma-linolenic acid and omega-3 fatty acids) or by reducing drug effectiveness through different mechanism (e.g. degradation of neomycine by sodium carboxymethylcellulose or absorption of nitrazepam in tablet dosage form by colloidal silicon) [36]. Excipients can also alternate (decrease or increase) permeability of intestinal membrane, and therefore affect bioavailability of the drug [37]. Due to age-related changes in metabolism, accumulation of excipients may occur in older patients, leading to toxicity and adverse effects. A list of such excipients are summarised in a review [38].”
Good job. I really think this paragraph improves your manuscript.
Reviewer 2 Report
After the manuscript has been modified, from my point of view the review has improved quality and understanding
Author Response
Thank you very much for all your comments and suggestions.